

# CircRNA: a rising star in leukemia

Qianan Li, Xinxin Ren, Ying Wang and Xiaoru Xin

College of Life Sciences, Zhejiang Normal University, Jinhua, China

## ABSTRACT

Non-coding RNA are a class of RNA that lack the potential to encode proteins. CircRNAs, generated by a post-splicing mechanism, are a newly discovered type of non-coding RNA with multi-functional covalent loop structures. CircRNAs may play an important role in the occurrence and progression of tumors. Research has shown that circRNAs are aberrantly expressed in various types of human cancers, including leukemia. In this review, we summarize the expression and function of circRNAs and their impact on different types of leukemia. We also illustrate the function of circRNAs on immune modulation and chemoresistance in leukemia and their impact on its diagnosis and prognosis. Herein, we provide an understanding of recent advances in research that highlight the importance of circRNAs in proliferation, apoptosis, migration, and autophagy in different types of leukemia. Furthermore, circRNAs make an indispensable difference in the modulation of the immunity and chemoresistance of leukemia. Increasing evidence suggests that circRNAs may play a vital role in the diagnostic and prognostic markers of leukemia because of their prominent properties. More detailed preclinical studies on circRNAs are needed to explore effective ways in which they can serve as biomarkers for the diagnosis and prognosis of leukemia *in vivo*.

# INTRODUCTION

Leukemia is a type of blood cancer derived from hematopoietic stem and progenitor cells (*Khodakarami et al., 2022*). These cells have lost the ability for self-renewal, differentiation, and apoptosis (*Milan et al., 2019*). Globally, 437,033 people were diagnosed with leukemia in 2018, making it one of the most common cancers (*Bispo, Pinheiro & Kobetz, 2020*). This disease caused 309,006 cancer deaths and ranked as the 11th leading cause of cancer death for both men and women according to GLOBOCAN 2018 data (*Bispo, Pinheiro & Kobetz, 2020*). Leukemia can be classified into four main categories, including acute myeloid leukemia (AML), acute lymphoblastic leukemia (ALL), chronic myeloid leukemia (CML), and chronic lymphocytic leukemia (CLL) (*Bhat et al., 2020*; *Deng, Chao & Zhu, 2023*; *Perez de Acha, Rossi & Gorospe, 2020*). AML is the most common form of acute leukemia found in adults (*Greiner, Gotz & Wais, 2022*; *Kang et al., 2022*; *Pinto-Merino et al., 2022*). It is an aggressive and heterogeneous disease characterized by rapid proliferation and arrested differentiation of medulloblastoma in the peripheral blood, bone marrow, and other tissues (*Shapoorian, Zalpoor & Ganjalikhani-Hakemi, 2021*; *Xiang et al., 2022*). ALL is a highly-aggressive hematological disease (*Li et al., 2022*; *Xin et al., 2022*). ALL patients

Corresponding author
Xiaoru Xin, xinxiaoru@zjnu.edu.cn

are characterized by an abnormal proliferation and aggregation of lymphocytes in the bone marrow. Lymphocytes are transferred to other tissues and organs, infiltrating systemic organs and tissues, and seriously affecting the hematopoietic function of the bone marrow (*Ruchel et al., 2022*; *Tan, Bertulfo & Sanda, 2017*). CML is a myeloproliferative neoplasm characterized by the proliferation of clonal hematopoietic stem cells (*Minciacchi, Kumar & Krause, 2021*; *Sampaio et al., 2021*). It is a relatively rare disorder with an annual incidence of 0.7–1 cases per 100,000 individuals with a lower incidence in females than in males (*Andretta et al., 2021*). CLL is the most common lymphoproliferative disease. The disease causes CD5+ B cells to proliferate and accumulate in the blood, spleen, lymph nodes, and bone marrow (*Hallek, 2019*; *Iovino & Shadman, 2020*; *Scheffold & Stilgenbauer, 2020*).

Circular RNAs (circRNAs) are an abundant class of endogenous noncoding RNAs (ncRNAs) that are characterized by covalently closed loop structures with no exposed 3′ and 5′ ends (*Chen & Shan, 2021*; *Kristensen et al., 2019*; *Zhou et al., 2020*). This structure was first described in viroids and were later found to be widely expressed in eukaryotic organisms (*Memczak et al., 2013*; *Wang et al., 2014*; *Wilusz & Sharp, 2013*). CircRNAs are divided into four types based on where they originate, namely exonic circRNAs, intronic circRNAs, exon-intron circRNAs, and intergenic circRNAs (*Chen et al., 2021*; *Wang et al., 2020*; *Wang & Ji, 2016*). CircRNAs have specialized structures and properties which allow them to be involved in various biogenesis processes (*Chen & Shan, 2021*; *Zhang, Hu & Yu, 2020a*). The literature reports that circRNAs participate in physiological and pathological processes in four unique ways (*Cui et al., 2020*; *Jamal et al., 2019*; *Wu, Li & Jin, 2019*). Firstly, it acts as a miRNA sponge (Fig. 1A). Increasing evidence suggests that circRNAs contain miRNA binding sites, and circRNAs can block miRNA binding to the 3′-UTR region of target mRNAs, thereby regulating the expression of mRNAs (*Jin et al., 2020*; *Li et al., 2020b*; *Rajappa et al., 2020*; *Zhang et al., 2020b*). Secondly, it regulates gene transcription. Most of the circRNAs exist in the cytoplasm, but a small fraction of circRNAs also exist in the nucleus, such as ciRNAs and ElciRNAs, suggesting that circRNAs may be involved in the transcriptional regulation of parental genes (Fig. 1B) (*Kumari et al., 2022*; *Li et al., 2020a*; *Wang et al., 2020*; *Zhou et al., 2022b*). CiRNAs and ElciRNAs regulate gene transcription differently. CiRNAs act mainly through their accumulation in the transcriptional region of the parental gene, enhancing the polymerase II (Pol II) extension activity to regulate transcription (*Li et al., 2020a*). However, EIciRNAs promote transcription mainly by interacting with the U1 small ribonucleic acid protein (snRNP) in the gene promoter region and binding to polymerase II (Pol II) (*Sun et al., 2020*). Thirdly, circRNA interaction with RBPs. CircRNAs can act on different proteins to form circular RNA-protein complexes (circRNPs) to regulate protein action, the subcellular localization of proteins or the expression of target genes (Fig. 1C) (*Qian et al., 2021*; *Singh et al., 2021*; *Sun et al., 2020*). Some circRNAs can act as a decoy to regulate gene expression by binding or dissociating proteins and some circRNAs are able to be used as protein scaffolds to mediate protein-protein interactions. Fourthly, accumulating evidence has shown that circRNAs can be translated into peptides or proteins, even though the noncoding properties of circRNAs are traditionally due to the absence of 5′ and 3′ ends (Fig. 1D) (*Pamudurti et al., 2017*; *Schneider & Bindereif, 2017*; *Yang et al., 2017*).

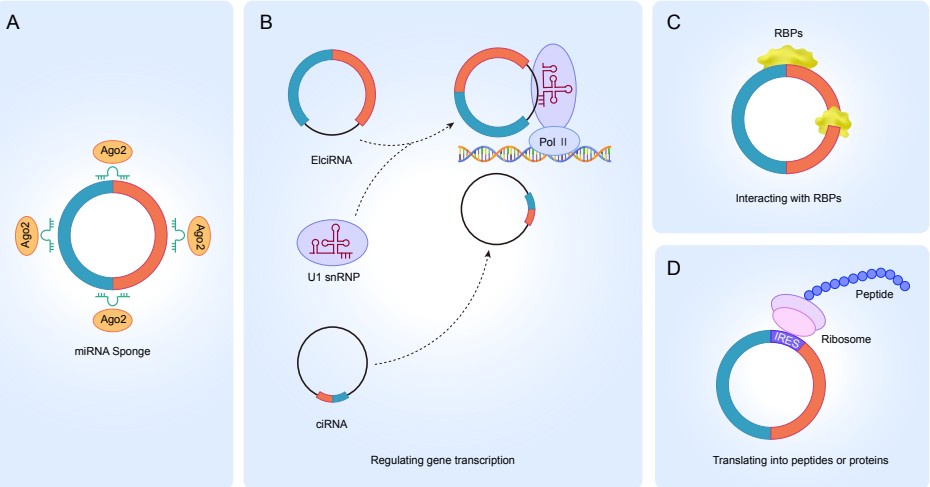

**Figure 1** **Functions of circRNA.** (A) circRNAs serve as an miRNA sponge to regulate the expression of target genes; (B) ElciRNAs bind to U1 snRNP to form a complex, which in turn binds to Pol II in the promoter region to regulate the transcriptional level of genes; ciRNAs can directly accumulate in the parental transcribed region to promote Pol II activity and regulate the expression of downstream genes; (C) circRNAs can interact with RBPs to regulate the function of related proteins; (D) circRNAs can translate into peptides or proteins.

The current research on circRNAs in leukemia is not comprehensive. Here, we briefly review the impact of the expression and function of circRNAs on different human leukemias, focusing on the role of circRNAs in leukemia immunomodulation and chemoresistance, as well as the potential of circRNAs in leukemia diagnosis and prognosis. This review help researchers to quickly understand the recent research progress of circRNAs in human leukemia and provide valuable information for future research directions.

## The intended audience and need for this review

CircRNAs are non-coding RNA with multiple functions that have a covalent-loop structure. Leukemia is a blood cancer derived from hematopoietic stem and progenitor cells. The current research on circRNAs in leukemia is not comprehensive. Therefore, we briefly reviewed the impact of the expression and function of circRNAs on different human leukemias, focusing on the role of circRNAs in leukemia immunomodulation and chemoresistance. We also focused on the potential of circRNAs in the diagnosis and prognosis of the disease as well as some issues faced by circRNAs in human leukemia. The preclinical studies of circRNAs need to be further developed to explore their use as diagnostic and prognostic biomarkers in leukemia in vivo. In summary, this review is meant to help researchers quickly understand the recent research progress concerning

the role of circRNAs in human leukemia and to provide valuable information for future research directions.

## SURVEY METHODOLOGY

To ensure an inclusive and unbiased analysis of the literature and to accomplish the review objectives, we searched the following literature databases: PubMed, Google Scholar and Web of Science. The search terms included: circRNA, leukemia, chemoresistance, immune modulation, and biomarker. It is worth noting that the keywords used and their variants and related words can be sorted, combined, and then searched.

  1. The terms searched for circRNA were: circRNA, circular RNA, noncoding RNA, circular intronic RNA, exon-intron circRNA, epigenetic, epigenetic regulation.

  2. The terms searched for leukemia were: leukemia, acute myeloid leukemia, acute myeloid leukemia, chronic myeloid leukemia, chronic lymphocytic leukemia, AML, ALL, CML, CLL.

  3. The terms searched for circRNA and leukemia were: chemoresistance, immune modulation, biomarker.

  This article is based on published literature. The aspects of the inclusion criteria are the retrieval keywords, information not covered by previous literature, and most importantly, the use of a clear and credible source.

### Abnormal circRNA expression and function in leukemia
#### circRNAs in acute myelogenous leukemia

Acute myelogenous leukemia (AML) is a highly aggressive malignancy of hematopoietic stem cells. A series of studies have shown that circRNAs are aberrantly expressed in AML (*Chang et al., 2022*; *Wu et al., 2021*). *Sun et al. (2019)* demonstrated that there was a trend of elevated expression of circMYBL2 in AML patients with an FLT3-ITD mutation. Mechanistically, circMYBL2 regulated downstream signaling pathways by upregulating the translation efficiency of FLT3 kinase, which in turn regulates AML proliferation and differentiation (*Sun et al., 2019*). *Liu et al. (2021)* found that circRNF220 accumulated in large amounts in the bone marrow of AML. In-depth studies revealed that the downregulation of circRNF220 could suppress AML cell proliferation and induce apoptosis of AML cells. Mechanistically, circRNF220 might act as a sponge for miR-30a to regulate the progression of AML (*Liu et al., 2021*). Additionally, hsa_circ_0003602 expression was upregulated in AML. Highly expressed hsa_circ_0003602 had a promotional effect on the proliferation, migration, and invasion of AML cells and inhibited AML apoptosis. Meanwhile, hsa_circ_0003602 may have a role in tumor progression in AML by binding to miR-502-5p and thus regulating the expression of IGF1R (*Ye et al., 2022*). *Hu et al. (2022)* found a  trend of upregulation of circ_KCNQ5 in pediatric AML patient cell lines. Their studies revealed that circ_KCNQ5 could target miR-622, which in turn inhibited the proliferation of AML cells and promoted apoptosis of AML cells. Conversely, there are some circRNAs that are downregulated in AML. For example, *Liu et al. (2022)* showed that circ_0004277 was downregulated in the bone marrow of AML. The overexpression

of circ_0004277 may significantly suppress cell viability, migration, and invasion in AML. Further studies illustrated that circ_0004277 impeded AML by binding to miR-134-5p and upregulating SSBP2 expression (*Liu et al., 2022*).

### circRNAs in acute lymphoblastic leukemia

Acute lymphoblastic leukemia (ALL) is a common type of malignancy that occurs in children. Growing evidence indicates that circRNAs may act as an oncogene, playing an important role in the development of ALL. It was reported that circPVT1 was significantly highly expressed in ALL patients, and knocking out circPVT1 could provoke apoptosis and suppresses the proliferation of ALL cells (*Hu et al., 2018*; *Jia & Gu, 2021*). Similarly, *Ling et al. (2021)* showed that circ-PRKDC was elevated in ALL patients. The downregulation of circ-PRKDC suppressed the RELN-mediated activation of PI3K/AKT/mTOR signaling pathway, which inhibited cell proliferation and enhanced the apoptosis and autophagy in ALL cells. In addition, circ_0000745 was significantly highly expressed in ALL patients and cell lines. Circ_0000745 could regulate NET1 expression by acting as a miR-494-3p sponge, which may affect the progression of ALL (*Yang et al., 2022*). However, circRNAs may be tumor suppressor genes in ALL. For example, *Zhu et al. (2021)* demonstrated that circADD2 was downregulated in the cells and tissues of ALL. CircADD2 could suppress ALL cell proliferation and promote ALL cell apoptosis by binding to miR-149-5p *in vitro* and *in vivo* (*Zhu et al., 2021*). Therefore, circRNAs may play oncogenic or anticancer roles in the progression of ALL. The function and mechanism of action of circRNAs must be further elucidated.

### circRNAs in chronic myelogenous leukemia

Chronic myelogenous leukemia (CML) is a myeloproliferative disease that originates primarily from hematopoietic stem cells. Studies have shown that a variety of circRNAs are aberrantly expressed in CML. *Feng et al. (2020)* found that circHIPK3 was highly expressed in the serum of CML patients. Further loss-of-function experiments showed that circHIPK3 played an oncogenic role in the development of CML (*Feng et al., 2020*). In addition, *Liu et al. (2018)* illustrated that the expression of hsa_circ_0080145 was aberrantly upregulated in CML compared with normal controls. In-depth studies demonstrated that hsa_circ_0080145 could sponge miR-29b to regulate the proliferation of CML cells. The knockdown of hsa_circ_0080145 was also shown to inhibit the proliferation of CML cells (*Liu et al., 2018*). Circ_100053 was also highly expressed in peripheral blood as well as in serum of patients with CML. The high expression of circ_100053 was closely related to the clinical stage of CML and the BCR/ABL mutation status (*Ping et al., 2019a*). circ_0009910 was also significantly upregulated in the serum of K562/R cells, and the up-regulation of circ_0009910 may promote the proliferation and autophagy of K562/R cells and suppress apoptosis of K562/R cells through miR-34a-5p/ULK1 axis (*Cao et al., 2020*).

### circRNAs in chronic lymphocytic leukemia

Chronic lymphocytic leukemia (CLL) is a clonal disease of mature B cells. Existing research has found that some circRNAs are expressed abnormally in CLL. *Wu et al. (2019)* found that

 

circ_0132266 was significantly downregulated in the mononuclear cells in the peripheral blood of CLL patients compared with normal individuals, and it could act as an endogenous sponge of hsa-miR-337-3p to regulate the expression of downstream genes. Mechanistically, circ_0132266 regulates the progression of CLL *via* the circ_0132266/miR-337-3p/PML axis, demonstrating that circ_0132266 was crucial for the malignant proliferation of CLL (Wu 2019). Circ-RPL15, a circRNA aberrantly expressed in CLL patients, was found to regulate the RAS/RAF1/MEK/ERK signaling pathway by binding to miR-146b-3p, affecting CLL progression (*Wu et al., 2020a*). *Xia et al. (2018)* determined that the knockdown of circ-CBFB significantly inhibited CLL cell proliferation, prevented the progression of the CLL cell cycle, and induced the apoptosis of CLL cells. Mechanistically, circ-CBFB sponged miR-607, which in turn activated the Wnt/ $\beta$-catenin signaling pathway, and facilitated the development of CLL (*Xia et al., 2018*). A mitochondrial genome-derived circRNA (mt-circRNAs), mc-COX2, was also found to be significantly overexpressed in the plasma exosomes of CLL patients. The upregulation of mc-COX2 accelerated cell proliferation and inhibited the apoptosis of CLL cells. Importantly, the high expression of mc-COX2 was positively correlated with worsening leukemogenesis and survival in CLL patients, suggesting that mc-COX2 made a crucial difference in the development and progression of CLL (*Wu et al., 2020b*).

## Role of circRNAs in chemoresistance in leukemia

As with many other cancers, drug resistance in leukemia patients is a difficult issue to address. Commonly used drugs for the treatment of leukemia, such as rituximab, chlorambucil, bendamustine, *etc.*, have obvious curative effects in the early stages of administration. Over time, however, the efficacy of these drugs declines significantly, and leukemia patients may develop resistance to these drugs. Growing evidence has suggested that circRNAs significantly impact the chemoresistance of leukemia patients (Fig. 2). *Cao et al. (2020)* demonstrated that Circ_0009910 was up-regulated in imatinib-resistant CML patient cells and it promoted imatinib-resistance by targeting miR-34a-5p to regulate ULK1 and induce autophagy. Similarly, circBA9.3 was shown to be upregulated in CML patients. CircBA9.3 could promote the resistance of CML cells to treatment with tyrosine kinase inhibitors (TKIs), suggesting that circBA9.3 might be a target for drug resistance in CML patients (*Pan et al., 2018*). *Ding et al. (2021)* found that circNPM1 could promote the resistance of AML cells to adriamycin *via* the miR-345-5p/FZD5 axis, indicating that circNPM1 was a potential marker in AML drug resistance therapy. circPAN3 was found to be abnormally highly expressed in doxorubicin-resistant AML cells, indicating that circPAN3 may promote AML resistance to doxorubicin by regulating cellular autophagy and affecting apoptosis-related proteins expression through the AMPK/mTOR signaling pathway (*Shang et al., 2019*). In addition, hsa_circ_0058493 was highly expressed in peripheral blood mononuclear cells (PBMCs) of CML patients, and its high expression was closely related to the poor clinical efficacy of imatinib. Therefore, hsa_circ_0058493 may be a new strategy for the treatment of CML (*Zhong et al., 2021*). *Ping et al. (2019a)* found that circ_100053 was significantly up-regulated in the serum of imatinib-resistant CML patients, and the expression of circ_100053 was correlated with its clinical stage. circ_100053 may be

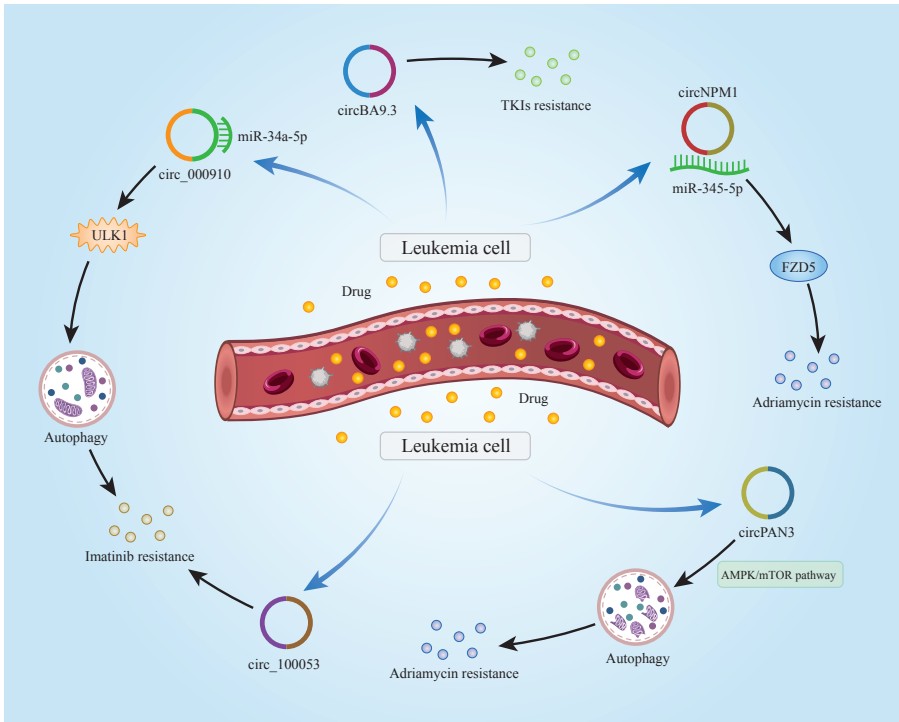

**Figure 2** **Mechanisms of circRNAs in leukemia chemoresistance.** CircRNAs are able to regulate chemoresistance in leukemia through multiple signaling pathways.

a potential biomarker of imatinib resistance in CML (*Ping et al., 2019a*). The above studies suggest that circRNAs do play an important role in chemoresistance in leukemia patients. It may be possible to use existing tools to develop relevant reagents for circRNAs that can treat chemoresistance in leukemia patients. In this way, the suffering caused by the disease may be alleviated in patients.

## Role of circRNAs in immune modulation in leukemia

Immune regulation means that the various immune cells and immune molecules of the body maintain their own physiologically dynamic balance and relative stability by promoting and restricting each other. Depending on the type of immune cells, the immune system can induce a series of responses to fight viruses, bacteria, and tumors. It is worth noting that circRNAs can be involved in some regulatory responses of the immune system. *Hou et al. (2021)* reported that circ_0000094 could target miR-223-3p to upregulate the expression of FBW7, thereby affecting the specific T cell proliferation, apoptosis, invasion, and migration in ALL, regulating a series of immune responses (*Hou et al., 2021*). *Feng, Li & Tang (2021)* showed that circ_0000745 was notably momre highly expressed in ALL. Circ_0000745 could sponge miR-193b-3p and prevent the interaction between miR-193b-3p and NOTCH1 mRNA, specifically regulating the proliferation and apoptosis of T cells in ALL (*Feng, Li*

& Tang, 2021). In addition, hsa_circ_0075001 was highly expressed in AML. The high expression of hsa_circ_0075001 was closely correlated with the low expression of genes related to the toll-like receptor signaling pathway. The toll-like receptor was an important component of the innate immune response, suggesting that hsa_circ_0075001 might be involved in the immune regulation of AML (Hirsch et al., 2017). Another circRNA, circMYBL2, was aberrantly expressed in AML patients with FLT3-ITD mutations. The downregulation of circMYBL2 significantly inhibited FLT3 kinase expression and its downstream FLT3-ITD signaling pathway, suggesting that circMYBL2 might be involved in the immune regulation of AML (Sun et al., 2019). It was reported that circRNA expression was found to be cell-specific in hematopoietic differentiation and this specificity increased as cell matured (Nicolet et al., 2018). In summary, circRNAs make a great deal difference in the modulation of leukemia immunity.

## CircRNAs as biomarkers for diagnosis and prognosis in leukemia

Recently, a number of circRNAs have been found to be aberrantly expressed in leukemia. A growing body of evidence points the use of circRNAs as effective biomarkers for the diagnosis and prognosis of leukemia due to their unique properties (Guo et al., 2022; Tayel et al., 2022; Zhou et al., 2022a). For example, the upregulation of circ_0009910 predicted poor diagnosis and prognosis in AML patients. Mechanically, the down-regulation of circ_0009910 expression could inhibit AML cells proliferation and induce AML cells apoptosis by sponging miR-20a-5p (Ping et al., 2019b). Besides, Circ-VIM is a circRNA of the VIM gene, and some studies have reported that overexpression of circ-VIM was associated with higher white blood cell (WBC) counts and shorter overall survival (OS) in AML. Therefore circ-VIM may be used as a prognostic predictor for AML patients (Yi et al., 2019). Wu et al. (2020a) showed that the upregulation of circ-RPL15 expression was positively associated with IGHV mutation status, an important factor for assessing CLL prognosis, indicating that circ-RPL15 might be a promising plasma marker for CLL diagnosis (Wu et al., 2020a). In addition, Zhou et al. (2019) used Kaplan–Meier analysis to determine that AML patients with circ-Foxo3 up-regulation lived longer than AML patients with circ-Foxo3 down-regulation. Patients with upregulated circ-Foxo3 after chemotherapy generally survived longer than those with downregulated circ-Foxo3. Recently, circRNF220 was determined to independently predict the prognosis of AML patients, and a high level of circRNF220 indicated ab unfavorable marker for relapse (Liu et al., 2021). These results demonstrated that circRNA could be an important biomarker for the prognosis of AML patients. The detection of circRNAs is an important task for clinical diagnosis and prognosis. CircRNAs can be detected using the following two methods: the first method is microarray-based circRNAs detection (Mi et al., 2022). In this method, total RNA is extracted from cells and linear RNA is removed by RNase R enzyme. The RNA is then amplified and transcribed using random primers, and the labeled circRNAs are hybridized to commercially available circRNAs. The hybridised microarray slides are cleaned and scanned using a fluorescence scanner, and the type and relative amount of circRNAs may be obtained from the fluorescence signal intensity. The second method is PCR-based circRNAs detection (Jinek & Doudna, 2009). The total RNA is first extracted

from the cells and then treated with RNase R to remove the linear RNA. The RNA was then reverse-transcribed to obtain complementary cDNA with circRNA. cDNA was subsequently amplified with specific primers and the fluorescent signal was measured and analysed.

## CONCLUSIONS AND PERSPECTIVES

At present, chemotherapy remains one of the most effective methods of treatment for leukemia. With the rapid development of technology, circRNAs have opened up a new research field as a novel type of non-coding RNA. Increasing evidence suggests that circRNAs can broadly regulate human physiological and pathological processes, especially in leukemia. In this review, we provided an overview of the latest research advances in circRNAs and highlighted the role they play in leukemia.

Increasing amounts of research have shown that circRNAs are aberrantly expressed in leukemia (Table 1). CircRNAs can serve as potential indicators or therapeutic targets for leukemia diagnosis and treatment. For example, circ_0009910 is upregulated in AML, and circ_0009910 can be used as a biomarker for its poor diagnosis and prognosis (*Ping et al., 2019b*). Meanwhile, circRNAs can act as oncogene or tumor suppressor genes to regulate the proliferation, apoptosis, cycle, differentiation, migration, invasion, autophagy, and other processes in leukemia, which play a crucial role in the occurrence and development of leukemia. In addition, circRNAs play a role in chemoresistance in leukemia. For example, circPAN3 regulates autophagy and affects the expression of apoptosis-related proteins through the AMPK/mTOR signaling pathway, resulting in AML resistance to doxorubicin (*Shang et al., 2019*). Furthermore, circRNAs are involved in the immune modulation of leukemia. For example, hsa_circ_0075001 can be involved in the toll-like receptor signaling pathway and participate in the immune modulation of AML (*Hirsch et al., 2017*). However, the research on circRNAs has been limited to *in vitro* experiments such as cells and tissues, and has not been applied to the clinical setting. Therefore, we need to increase our preclinical investment to explore effective methods of applying circRNAs as biomarkers for the diagnosis and prognosis in leukemia in vivo. First, more *in vitro* experiments should be conducted to screen for efficient and expected circRNAs. Second, a robust circRNA database should be established, especially those with tissue specificity. Third, novel tools and methods should be developed to improve the specificity of circRNA detection. In conclusion, circRNAs are a class of regulatory factors that can play an important role in leukemia, and they can regulate the occurrence and development of leukemia. Therefore, it is necessary to strengthen the research on circRNAs in the future so that the results can be transferred from *in vitro* research to *in vivo* research. These applications will better serve those affected by leukemia and will allow for the development of precision medicine in the near future.

Peer

**Table 1  CircRNAs in leukemia.**

| Type of leukemia | circRNAs | Expression | miRNAs, RBPs, and pathways targeted | Function | Sample | Ref. |
|---|---|---|---|---|---|---|
| AML | circMYBL2 | UP | FLT3 | Promotes proliferation and inhibites differentiation | Human FLT3-ITD cell lines and AML primary samples | *Sun et al. (2019)* |
| AML | circRNF220[*] | UP | miR-30a | Increases proliferation and suppresses apoptosis; poor prognosis | Patients | *Liu et al. (2021)* |
| AML | hsa_circ_0003602 | UP | miR-502-5p and IGF1R | Promotes proliferation, migration and invasion and inhibits apoptosis | Patients | *Ye et al. (2022)* |
| AML | circ_KCNQ5 | UP | miR-622 and RAB10 | Inhibits proliferation and promotes apoptosis | Patients | *Hu et al. (2022)* |
| AML | circ_0004277 | DOWN | miR-134-5p and SSBP2 | Promotes cell viability, migration and invasion | Human bone marrow stromal cell line and human AML cell lines | *Liu et al. (2022)* |
| AML | circNPM1 | UP | miR-345-5p and FZD5 | Strengthens Adriamycin resistance | Patients | *Ding et al. (2021)* |
| AML | circPAN3 | UP | AMPK/mTOR signaling patway | Promotes resistance to doxorubicin | Human AML cell lines | *Shang et al. (2019)* |
| AML | hsa_circ_0075001 | UP | Toll-like receptor signaling pathway | Correlates with Toll-like receptor signaling pathway | Patients | *Hirsch et al. (2017)* |
| AML | circ_0009910[*] | UP | miR-20a-5p | Increases proliferation and inhibits apoptosis; poor diagnosis and prognosis | Patients | *Ping et al. (2019a)*; *Ping et al. (2019b)* |
| AML | circ-VIM[*] | UP | Unknown | Promotes occurrence and progression; poor prognosis | Patients | *Yi et al. (2019)* |
| AML | circ-Foxo3[*] | DOWN | Apoptotic pathways | Poor prognosis | Patients | *Zhou et al. (2019)* |
| ALL | circPVT1 | UP | Notch signaling pathway through miR-30e/DLL4 | Promotes proliferation and induces apoptosis | Patients | *Jia & Gu (2021)* |

**Table 1** (*continued*)

| Type of leukemia | circRNAs | Expression | miRNAs, RBPs, and pathways targeted | Function | Sample | Ref. |
|---|---|---|---|---|---|---|
| ALL | circ-PRKDC | UP | PI3K/AKT/mTOR signaling pathway through miR-653-5p/Reelin | Increases cell proliferation and suppresses apoptosis and autophagy | Human T-ALL cell lines and peripheral blood mononuclear cells | *Ling et al. (2021)* |
| ALL | circ_0000745 | UP | Notch signaling pathway through miR-193b-3p | Promotes proliferation and suppresses apoptosis | Patients | *Feng, Li & Tang (2021)*; *Yang et al. (2022)* |
| ALL | circADD2 | DOWN | miR-149-5p and AKT2 | Suppresses proliferation and promotes apoptosis | Patients | *Zhu et al. (2021)* |
| ALL | circ_0000094 | DOWN | miR-223-3p and FBW7 | Enhances proliferation, migration, and invasion and inhibits apoptosis | Patients | *Hou et al. (2021)* |
| CML | circHIPK3 | UP | Unknown | Promotes progression | Patients | *Feng et al. (2020)* |
| CML | hsa_circ_0080145 | UP | miR-29b | Promotes proliferation | Mesenchymal stem cell lines | *Liu et al. (2018)* |
| CML | circ_100053 | UP | Unknown | Strengthens imatinib resistance | Patients | *Ping et al. (2019a)*; *Ping et al. (2019b)* |
| CML | circ_0009910 | UP | miR-34a-5p and ULK1 | Promotes proliferation and autophagy and suppresses apoptosis; promotes imatinib-resistance | Patients | *Cao et al. (2020)* |
| CML | circBA9.3 | UP | cABL1 and BCR-ABL1 | Promotes resistance to TKIs | Patients | *Pan et al. (2018)* |
| CML | hsa_circ_0058493 | UP | miR-548b-3p | Inhibits resistance to imatinib | Patients | *Zhong et al. (2021)* |
| CLL | circ_0132266 | DOWN | miR-337- 3p and PML | Promotes cell viability | Patients | *Wu et al. (2019)* |
| CLL | circ-RPL15[*] | UP | miR-146b-3p | Promotes progression; poor prognosis | Patients | *Wu et al. (2020a)*; *Wu et al. (2020b)* |
| CLL | circ-CBFB | UP | Wnt/ $\beta$-catenin pathway through miR-607/FZD3 | Promotes proliferation, cell cycle progression and inhibits apoptosis | Patients | *Xia et al. (2018)* |
| CLL | mc-COX2 | UP | Unknown | Increases proliferation and inhibits apoptosis | Patients | *Wu et al. (2020a)*; *Wu et al. (2020b)* |

**Notes.**
    *Indicates the circRNAs used for diagnosis and prognosis.

Li et al. (2023), *PeerJ*, DOI 10.7717/peerj.15577

### Funding

This work was supported by the Natural Science Foundation of Zhejiang Province (LQ21C060003), and the Jinhua Science and Technology Bureau (2021-3-148). The funders had no role in study design, data collection and analysis, decision to publish, or preparation of the manuscript.

### Grant Disclosures

The following grant information was disclosed by the authors:
Natural Science Foundation of Zhejiang Province: LQ21C060003.
Jinhua Science and Technology Bureau: 2021-3-148.

### Competing Interests

The authors declare there are no competing interests.

### Author Contributions

- Qianan Li conceived and designed the experiments, performed the experiments, analyzed the data, prepared figures and/or tables, authored or reviewed drafts of the article, and approved the final draft.
- Xinxin Ren performed the experiments, authored or reviewed drafts of the article, and approved the final draft.
- Ying Wang performed the experiments, authored or reviewed drafts of the article, and approved the final draft.
- Xiaoru Xin conceived and designed the experiments, performed the experiments, prepared figures and/or tables, authored or reviewed drafts of the article, and approved the final draft.

### Data Availability

This study is a literature review.

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
