# Peer review of "CircRNA: a rising star in leukemia"

_PeerJ, doi:10.7717/peerj.15577_

## Round 0.1 · original submission · Major Revisions

This manuscript has been reviewed by three reviewers. Although all reviewers agreed that this manuscript is an interesting article, they have suggested many changes to the article before it can be accepted for publication. Please revise the manuscript accordingly and submit it at the earliest.

Look forward to receiving the revised manuscript soon.

Reviewer 1 ·

Basic reporting

Li et al. summarized the circRNAs related to leukemia in this short review manuscript. The authors start with an introduction of leukemia and circRNAs and then illustrate the abnormal circRNAs in four different classes of leukemia. In the following, the related circRNAs are extended to chemoresistance, immune modulation, and biomarkers for diagnosis and prognosis in leukemia. The authors listed several examples for each of these classifications and described briefly how these circRNAs related to leukemia.
1. This review could be interesting for the researcher in ncRNA and leukemia. However, there has been a couple of recent reviews on the topic that cover most of the aspects discussed here (doi.org/10.1002/iid3.725 and doi.org/10.1002/JLB.2RU0619-213R). To make the manuscript unique, the authors could extend more on those circRNAs not specifically classified and discussed in the previous reviews (it is a good point to extend to chemoresistance, immune modulation, and biomarkers for diagnosis and prognosis in leukemia). They could also add a speculative section or some data integration analysis that would move the field further ahead.
2. The circRNAs introduction parts should be merged into the introduction section and only keep the most important information. In the current version, there are too much information that are not related to the main topic of the review in this part. Especially there are many examples of how circRNAs participate in gene regulations, however, most of those circRNAs (ciRS-7, circ-ANKRD52, circ-EIF3J, circ-PAIP2, circPABPN1, and circ-SHPRH) have nothing to do with the main topic of leukemia.
3. Could extend more information to the circRNAs detection techniques, which could be the limitation to applying the circRNAs to clinical diagnosis and prognosis.
4. The content of Table 1 and Figure 2 are highly overlapped. I suggest only keeping Table 1 and removing Figure 2. In addition, Table 1 could include more information such as the miRNA or protein targets.
5. It is better to highlight the potential circRNAs for diagnosis and prognosis in Table 1. Or add an extra figure for the circRNAs for diagnosis and prognosis.
6. The English language needs better polished. Many expressions could be improved to a more professional, unambiguous style.

Experimental design

The Survey Methodology looks good with comprehensive, unbiased coverage of the subject. The references are correctly cited.
The manuscript should be better organized as mentioned in the basic reporting. Please note the introduction parts and the tables and figures.

Validity of the findings

It’s better to add a prospective part on the clinical applications of circRNAs in leukemia, including the feasibility analysis on the potential diagnostic and therapeutic circRNA, the limitation of the application of circRNA, and the technical issues for clinical trials. This could help further point out the future direction of this area.

Reviewer 2 ·

Basic reporting

In this review manuscript, the authors provided some summaries of the expression and function of circRNAs and their impact on different types of leukemia and emphasized the function of circRNAs on immune modulation and chemoresistance in leukemia and the potential of circRNAs on diagnosis and prognosis in different types of leukemia.

The language used in this manuscript is clear, unambiguous, and professional.

The abbreviations and background information are clearly mentioned. However, in the manuscript, a lot of different circRNAs are mentioned, it would be better if the mechanism of those circRNAs are supplemented in Fig. 1 by adding the circRNA names into the figure.

The author even made a section of "why this review is needed and who it is intended for" to claim their motivation for the review.

Fig.1 can be improved by adding the names of mentioned circRNAs to the figure.

Experimental design

The study design is described with sufficient detail in the Survey Methodology part.

The sources are adequately cited.

The review is organized logically into coherent paragraphs and subsections.

Validity of the findings

The authors pointed out the importance of circRNA in the leukemia field and updated the readers with recent discoveries.

Reviewer 3 ·

Basic reporting

The review titled 'CircRNA: a rising star in leukemia' is a well written review on circular RNAs in context of leukemia.
I have the following few suggestions for the authors for the improvement of the manuscript:
1. Please include a figure showing biogenesis of circRNAs.
2. Please include the following reference in the introduction section:
- Rajappa A, Banerjee S, Sharma V, Khandelia P. Circular RNAs: Emerging Role in Cancer Diagnostics and Therapeutics. Front Mol Biosci. 2020 Oct 28;7:577938. doi: 10.3389/fmolb.2020.577938. PMID: 33195421; PMCID: PMC7655967.
- Deng W, Chao R, Zhu S. Emerging roles of circRNAs in leukemia and the clinical prospects: An update. Immun Inflamm Dis. 2023 Jan;11(1):e725. doi: 10.1002/iid3.725. PMID: 36705414; PMCID: PMC9801069.
- Zhou M, Gao X, Zheng X, Luo J. Functions and clinical significance of circular RNAs in acute myeloid leukemia. Front Pharmacol. 2022 Nov 24;13:1010579. doi: 10.3389/fphar.2022.1010579. PMID: 36506538; PMCID: PMC9729264.
- Singh V, Uddin MH, Zonder JA, Azmi AS, Balasubramanian SK. Circular RNAs in acute myeloid leukemia. Mol Cancer. 2021 Nov 18;20(1):149. doi: 10.1186/s12943-021-01446-z. PMID: 34794438; PMCID: PMC8600814.
3. Table 1: make an extra column and mention whether the study was done with cell line (name?) or patient sample or mice?

Experimental design

No comment.

Validity of the findings

No comment.

---

## Round 0.2 · accepted · Accept

The authors have addressed all the concerns raised by the reviewers and is ready for publication.

Reviewer 1 ·

Basic reporting

The manuscript is improved dramatically after revision. The authors fully addressed my concerns. The current version is suitable to publish on peerj.

Experimental design

No further comments

Validity of the findings

No further comments

Reviewer 3 ·

Basic reporting

Authors have made the corrections suggested to my satisfaction. The manuscript may be accepted.

Experimental design

No comment.

Validity of the findings

No comment.